# Comparison of Immune-Related Gene Expression in Two Chicken Breeds Following Infectious Bronchitis Virus Vaccination

**DOI:** 10.3390/ani13101642

**Published:** 2023-05-15

**Authors:** Schwann Chuwatthanakhajorn, Chi-Sheng Chang, Kannan Ganapathy, Pin-Chi Tang, Chih-Feng Chen

**Affiliations:** 1Department of Animal Science, National Chung Hsing University, Taichung 402, Taiwan; schwann.chu@mahidol.ac.th; 2Faculty of Veterinary Science, Mahidol University, Salaya, Nakhon Pathom 73170, Thailand; 3Department of Animal Science, Chinese Culture University, Taipei 111, Taiwan; 4Institute of Infection, Veterinary & Ecological Sciences (IVES), University of Liverpool, Neston CH64 7TE, UK; k.ganapathy@liverpool.ac.uk; 5The iEGG and Animal Biotechnology Center, National Chung Hsing University, Taichung 402, Taiwan; 6Smart Sustainable New Agriculture Research Center (SMARTer), Taichung 402, Taiwan

**Keywords:** RNA-seq, infectious bronchitis virus vaccination, spleen transcriptomic, immune response traits, differentially expressed gene

## Abstract

**Simple Summary:**

Host genetics plays a significant role in the effectiveness of immune responses against pathogens. Disease severity for an individual or population is associated with dissimilarity in the levels of host gene expressions. In this study, the variations in the effects of host immune responsiveness between Taiwan Country and White Leghorn chicken breeds were investigated by next-generation sequencing. Overall, immune response-related genes between Taiwan Country chicken and White Leghorn chicken were expressed differently against live attenuated infectious bronchitis virus vaccination. The major histocompatibility complexes and cytokines were determined as significant players in determining the pattern and magnitude of immune responses following an infection. This study demonstrated the host genetic influences on the development of adaptive immune response between the Taiwan Country and the White Leghorn chickens after infectious bronchitis virus vaccination.

**Abstract:**

This study aims to identify the immune-related genes and the corresponding biological pathways following infectious bronchitis virus vaccination in Taiwan Country and White Leghorn chicken breeds. Transcriptomic analyses of the spleen of these two breeds were conducted by next-generation sequencing. Compared to White Leghorn chicken, Taiwan Country chicken showed a significantly higher level of anti-infectious bronchitis virus (IBV) antibodies at 14 and 21 days pos vaccination. At 7 days post vaccination, in the Taiwan Country chicken, higher expression of mitogen-activated protein kinase 10, Major histocompatibility complex class 1, and V-set pre-B cell surrogate light chain 3 were found. In contrast, the White Leghorn chicken had a high expression of interleukin 4 induced 1, interleukin 6, and interleukin 22 receptor subunit alpha 2. These findings have highlighted the variations in immune induction between chickens with distinct genetic background and provided biological pathways and specific genes involved in immune responses against live attenuated IBV vaccine.

## 1. Introduction

Infectious bronchitis (IB) is a highly contagious disease in chickens, substantially affecting the poultry industry worldwide. This disease is caused by avian corona infectious bronchitis virus, a member of Gammacoronavirus [1]. Most clinical signs are related to the upper respiratory, urinary, and reproductive systems [2]. The mortality rate of IB infection ranges from 0% to 82%, depending on the viral strain, health status, and age of birds [3]. The virus primarily replicates in the trachea and causes ciliary stasis, leading to secondary bacterial infection, such as *Escherichia coli* and *Mycoplasma gallisepticum* infection [4]. This results in respiratory distress and poor growth performance [5]. In addition, the disease can cause significant loss due to a drop in egg production and poor egg quality [6,7].

Several studies have noted the influence of host genetic background on the immune response following inoculation of virulent or avirulent (vaccine) poultry viruses. Different breeds and sublines showed the variation of genes expression following infection, affecting the level of susceptibility to diseases [8,9,10,11]. The polymorphism of MHC genes was found to be particularly associated with resistant traits against pathogenic bacteria and viruses [12,13]. In addition, several genes were reported as disease-resistant genes in chicken, including Cyclophilin B (PPIB), MX Dynamin Like GTPase 1 (MX1), and 2’-5’-Oligoadenylate Synthetase Like (OASL) [14,15]. Vaccination studies also revealed the direct effect of host genetic background on variation of innate, mucosal, cellular, and humoral immune responses [16,17,18,19].

The spleen is an important organ in chickens, especially for development of both cellular and humoral immune responses [20]. The white pulp of the spleen contains ellipsoids, peri-ellipsoid sheaths (PESS), and lymph nodes, which are the main sites for initiating the humoral immune response by activating and differentiating B cells against blood-borne antigens [21]. This process leads to the secretion of large amounts of antibodies into the bloodstream and the development of memory B cells [22]. T lymphocytes are responsible for cell-mediated immune responses, and their activation occurs in the periarteriolar lymphoid sheaths (PELS), which allow for the monoclonal expansion and differentiation of naive T cells into effector T cells [23]. Gaining an understanding of the biological mechanisms underlying gene expression in the spleen could provide valuable insights into the effects of genetics on the development of immune responses in chickens.

Taiwan Country chicken (TCC) is a native breed in Taiwan, which was selected over 30 generations at National Chung Hsing university for prodigious characteristics including high growth rate and feasible adaptation. On the other hand, White Leghorn chicken (WLC) is a Mediterranean breed with high rate of egg production and outstanding feed efficiency. Previous studies have reported a variation in susceptibility between TCC and WLC responding to infectious diseases such as Newcastle disease, Marek’s disease, and Leucocytozoonosis [24,25,26]. The use and comparative analysis of these two genetically distinct breeds of chickens have the potential to elucidate the host genetic components that are associated with differential susceptibility to infections.

The aim of this study is to enhance our knowledge on the influence of host genetics on immune responses after IBV vaccination. RNA sequencing was used to cross-compare the immune-related genes of IBV-vaccinated Taiwan Country chickens (TCCs) and White Leghorn chickens (WLCs). The findings of this study would increase the availability of effective disease control strategies.

## 2. Materials and Methods

### 2.1. Animals and Tissue Sampling

In this experiment (Figure 1), B strain TCCs and WLCs were obtained from the National Chung Hsing University and Taiwan Hubbard GP Farm, respectively. Thirty-six day-old chicks of each type were obtained and raised in a disease-free facility with *ad libitum* feed and water. At 3 weeks old, the chickens of each breed were separated into two groups (18 each in the control and IBV groups). The control group received an IBV-free diluent intranasally, whereas the IBV group received live attenuated IBV, variant strain 4-91 (Nobilis IB 4-91, 7.2 log_10_ EID_50_, MSD Animal Health, Boxmeer, The Netherlands) intranasally.

Nine birds from each group of TCC and WLC were bled through the wing vein at 0 (before inoculation), 7, 14, and 21 days after the sham or IBV inoculations. After 2 h, the blood was centrifuged at 10,000 rpm for 10 min, and separated sera were stored in a freezer at −20 °C. At 7 days post vaccination, nine chickens of each group were humanely killed through cervical dislocation and spleens were collected. Each sample was transferred into a microcentrifuge tube filled with tissue storage reagent (RNAlater, Sigma-Aldrich, Singapore), and then directly frozen in a freezer at −80 °C.

### 2.2. Enzyme-Linked Immunosorbent Assay

All serum samples were tested for the antibody titer response to IBV by using the IB enzyme-linked immunosorbent assay kit (IDEXX IBV, Invitrogen, Taipei, Taiwan) following the manufacturer’s protocol. The optical density (OD) values were measured using a microplate spectrophotometer (Epoch, BioTek, Taipei, Taiwan). Antibody titers were determined following the manufacturer’s standard protocol using the sample-to-positive (S/P) ratio method and were reported with a positive cut-off value of 0.2.

### 2.3. RNA Extraction and cDNA Synthesis

Fifty milligrams of spleen from each sample was used for RNA extraction using total ribonucleic acid (RNA) isolation reagent (TRIzol Reagent, Invitrogen, Taipei, Taiwan) following the manufacturer’s protocol. The quality and concentration were determined using the microplate spectrophotometer (Epoch, BioTek, Taipei, Taiwan). Nine total RNA samples from each group were randomly pooled into three samples of equal amounts (three RNA pools in each of the three samples). In addition to using the total RNA for RNA sequencing, some RNAs were converted into complementary deoxyribonucleic acid (cDNA) by using the cDNA synthesis kit (RevertAid First Strand cDNA Synthesis kit, Thermo Scientific, Taipei, Taiwan) following the recommended protocol for quantitative real-time polymerase chain reaction (qPCR).

### 2.4. Library Construction and RNA Sequencing

RNAs extracted from spleen samples were analyzed through RNA Sequencing (RNA-Seq) with NGS, which was conducted by Biotools Co., Ltd. (New Taipei, Taiwan). RNA purity and quantification were checked using SimpliNano-Biochrom Spectrophotometers (Biochrom, Holliston, MA, USA). RNA degradation and integrity were monitored using Qsep 100 DNA/RNA Analyzer (BiOptic Inc., New Taipei, Taiwan). In total, 1 μL of total RNA per sample was used as input materials for RNA sample preparation. Sequencing libraries were generated using the KAPA mRNA HyperPrep Kit (KAPA Biosystems, Roche, Basel, Switzerland) following the manufacturer’s recommendations, and index codes were added to the attribute sequences of each sample. Briefly, messenger RNA (mRNA) was purified from the total RNA by using magnetic oligo-dT beads. Captured mRNA was fragmented through incubation at a high temperature in the presence of magnesium in the KAPA Fragment, Prime, and Elute Buffer. The first strand of cDNA was synthesized using random hexamer priming. A combination of second strand synthesis and A-tailing converted the cDNA–RNA hybrid into double-stranded cDNA (dscDNA), incorporated deoxyuridine triphosphate (dUTP) into the second cDNA strand, and added dAMP to the 3′ ends of the resulting dscDNA. The dsDNA adapter with 3′-dTMP overhangs were ligated to library insert fragments to generate library fragments carrying the adapters. To select cDNA fragments of 300–400 base pairs (bp), the library fragments were purified using the KAPA Pure Beads system (KAPA Biosystems, Roche, Basel, Switzerland). The library carrying the appropriate adapter sequences at both ends was amplified using KAPA HiFi HotStart ReadyMix (KAPA Biosystems, Roche, Basel, Switzerland) and library amplification primers. The strand marked with dUTP was not amplified, allowing strand-specific sequencing. Finally, PCR products were purified using the KAPA Pure Beads system, and library quality was assessed using the Qsep 100 DNA/RNA Analyzer (BiOptic Inc., New Taipei, Taiwan). All libraries were sequenced using the NovaSeq 6000 platform for paired-end 150 bp sequencing.

### 2.5. RNA-Seq Data Analysis and Go and KEGG Enrichment Analysis

In this study, the low-quality regions in the reads were removed using Trimmomatic based on the following criteria: THREADS:4; PHRED:33; ILLUMINACLIP: TruSeq3-PE. fa: 2:30:10; LEADING:3; TRAILING:3; SLIDINGWINDOW: 4:20 and MINLEN:36 [27]. The output files were further analyzed using FastQC [28] and MultiQC [29] to ensure the quality of the data. Clean reads were mapped to the *Gallus gallus* reference genome (GRCg7b) obtained from the National Center for Biotechnology Information by using HISAT2 software. Prior to the differential expression (DE) analysis, low-expression genes were excluded. Then, DE analysis between groups was performed using DESeq2 [30]. The *p*-values were corrected using the Benjamini–Hochberg procedure to control the false discovery rate (FDR). Transcripts that passed the cut-off criteria of |log2 (FoldChange)| > 1 and with a *p*-value < 0.05 were considered as differentially expressed genes (DEGs). Candidate genes were used to assess the biological pathway by using the Gene Ontology (GO) database and Kyoto Encyclopedia of Genes and Genomes (KEGG) database.

### 2.6. Quantitative Real-Time PCR Verification

The RNA-Seq results were validated through qPCR analysis with six selected genes, namely, interleukin 6 (IL-6), interleukin 22 (IL-22), C-C motif chemokine ligand 19 (CCL-19), C-X-C motif chemokine receptor 4 (CXCR4), C-X-C chemokine ligand 13-like 2 (CXCL13L2), and CD 34 molecule (CD34). These immune-related genes were selected based on their involvement in the GO and KEGG pathways and the availability of the sequences in the National Center for Biotechnology Information (NCBI) database. qPCR was performed using SYBR Green qPCR assays. The total volume of the reaction mixture was 10 µL, which comprised 5 µL of qPCR Master mix (PowerUP SYBR Green Master Mix, Appliedbiosystems, Taipei, Taiwan), 1 µL of cDNA from spleen extract, 3.8 µL of nuclease-free water, and 0.2 µL of forward and reverse primers (Appendix A). Then, each sample was placed into a qPCR analyzer (Step One Plus, Appliedbio systems, Taipei, Taiwan). In the analysis step, Step one software version 2.3 was used, followed by a qPCR cycle consisting of a Fast cycling mode (Primer Tm 60.0 °C) involving 40 cycles of denaturation (95.0 °C for 15 s), annealing (60.0 °C for 30 s), and elongation (60.0 °C for 30 s). In this study, we selected glyceraldehyde-3-phosphate dehydrogenase (GAPDH) as the endogenous gene due to its stable expression in our samples, and all qPCR reactions were performed in triplicate. Relative mRNA expression was calculated using the 2^−ΔΔCt^ method [31].

### 2.7. Statistical Analysis

The differences in antibody titers against IBV between treatment and control groups in each breed were analyzed by *t*-test using the TTEST procedure. All statistical analyses were conducted using SAS software (SAS, 2012).

## 3. Results

### 3.1. Anti-IBV ELISA Titers

In this study, following the vaccination of the 3-week-old birds, there were no clinical signs. The mean anti-IBV ELISA titers in each breed of birds is presented in Table 1. Compared to WLCs, TCCs showed earlier and higher antibody production. Compared with the control group, the antibody titers of the vaccinated TCC and WLC were significantly higher at 7, 14, and 21 or 14 and 21 days post vaccination, respectively. When comparing the IBV-inoculated TCC and WLC birds, significantly higher antibody titers were observed in TCCs at 14 and 21 days post vaccination.

### 3.2. Splenic Transcriptome Sequencing

The results of all 12 splenic transcription data (68.12 Gb), with the GC content at approximately 44–49% and the percentage of Q30 at >93% (Appendix A), indicated that the data could be used for further analysis. In addition, the comparison efficiency of the total reads compared with the GRCg7b reference genome of the 12 samples was 93–95% (Appendix A), and the comparison efficiency of the percentage of read and the reference genome was approximately 87–93%.

### 3.3. Differentially Expressed Gene Profiling

To identify the effect of genetics in IBV vaccination, vaccinated and unvaccinated chickens of the same breed were compared (Table 2). In TCCs, 257 upregulated and 186 downregulated genes were found. Upregulated significant immune-related genes involved in the early immune response and B-cell activation were macrophage mannose receptor 1-like 2, mitogen-activated protein kinase 10 (MAPK10), transforming growth factor beta receptor 3, immunoglobulin superfamily DCC subclass member 4, MHC, and class I and V-set pre-B cell surrogate light chain 3. In contrast, WLCs showed 49 and 72 upregulated and downregulated genes, respectively. The upregulated genes in response to vaccination were the genes for interleukin 4 induced 1 (IL-4I1), IL-6, and interleukin 22 receptor subunit alpha 2 (IL22RA2).

In the unvaccinated groups between breeds (Table 3), 577 and 1118 genes were up-regulated and downregulated, respectively. TCCs highly expressed IL-4I1, IL-6, IL-22, C-X-C, motif chemokine ligand 14, and leukocyte immunoglobulin-like receptor subfamily B member 5. Upregulated genes in WLCs were the genes for interleukin 1 receptor-like 2, MAPK10, T cell-interacting, activating receptor on myeloid cells protein 1-like, and tumor necrosis factor (TNF) receptor superfamily member 19.

Regarding the vaccinated groups between breeds (Table 3), 470 and 416 genes were upregulated and downregulated, respectively. Immune-related genes highly expressed in TCCs were IL-8L1, IgG Fc-binding protein-like, c-type lectin domain family 2 member L-like, class I histocompatibility antigen F10 alpha chain-like, and MHC class II beta chain BLB1. In WLCs, highly upregulated genes were IL-6, IL-22 receptor subunit alpha 2, T cell surface glycoprotein CD8 alpha chain-like, granzyme K, MHC class I antigen YF5, and MHC class II beta chain BLB2.

### 3.4. GO and KEGG Databases of DEGs

To assess crucial biological processes, the pathways were analyzed according to GO and KEGG database. Significant pathways in each comparison from TCCs and WLCs are shown in Appendix A.

Regarding the comparison between treatment and control groups, the GO database of TCCs (Appendix A) revealed significant differences in terms of response to chemicals, response to hormones, and taxis. In addition, differences were found in terms of cytokine–cytokine receptor interaction, cytosolic DNA-sensing pathway, and influenza A in the KEGG pathway analysis (Appendix A).

On the other hand, the GO pathway analysis of WLCs (Appendix A) indicated the enriched pathway in term of IL-6 production, positive regulation of T cell proliferation, and positive regulation of B cell activation. Likewise, the KEGG pathway analysis (Appendix A) showed that the significant enriched pathways responding to vaccination in WLCs were those related to neuroactive ligand–receptor interaction, taurine and hypotaurine metabolism, phenylalanine metabolism, and cytokine–cytokine receptor interaction.

Comparing the vaccinated groups of the two breeds (Table 4), the significant GO enriched pathways in TCCs were those related to cell–cell adhesion through plasma-membrane adhesion molecules, cell differentiation, chemotaxis, and regulation of leukocyte migration. Meanwhile, significant GO pathways of WLCs were those involving cell proliferation, acute inflammatory response, cell differentiation, regulation of cell proliferation, and positive regulation of cell adhesion. From the KEGG pathway analysis (Table 5), B strain TCCs and WLCs were those related to cell adhesion molecules (CAMs), cytokine–cytokine receptor interaction, toll-like receptor (TLR) signaling pathway, salmonella infection, and influenza A.

### 3.5. Validation Based on qPCR

In this study, significantly, DEGs related to innate and adaptive immune responses, including IL-6, IL-22, CCL-19, CXCR4, CXCL13L2, and CD34, were selected by using qPCR assay to confirm the creditability of the RNA-sequencing technique. The expression patterns of all DEGs from qPCR analysis were concomitant with those from RNA-Seq data analysis (Figure 2).

## 4. Discussion

Genetics play a crucial role in immune responses to infectious pathogens and vaccines [32]. Vigorous innate immune responses, rates of macrophage differentiation and activation, and MHC haplotype were well recognized as essential factors associated with disease resistance and susceptibility [33]. Between the breeds, variations in the MHC haplotype and immune-related genes define the outcomes of immune responses and survival ability. In this experiment, the comparison of transcriptomics between distinct genetic line chickens could provide better understanding on transcriptional responses in particular breeds and it revealed significant immune-related genes and molecular mechanisms against IBV vaccination between vaccinated and control groups, as well as between TCC and WLC.

### 4.1. Immune Responsiveness in Taiwan Country Chickens

From week 2 post vaccination, vaccinated TCCs showed higher antibody levels compared with the controls within 1 week after vaccination. At the same time, with the rise of systemic anti-IBV immunoglobulin, numerous genes involved in innate immune responses, B-cell accumulation and activation were highly expressed. The significant upregulated genes in TCCs responding to immunization were MAPK10, macrophage mannose receptor 1-like 2, Gallus gallus immunoglobulin-like receptor CHIR-B2-like, and V-set pre-B cell surrogate light chain 3. The Gallus gallus immunoglobulin-like receptor CHIR-B2-like is mainly expressed on B cells and its main function is inhibitory receptor-related with B cell proliferation process [34]. In addition, V-set pre-B cell surrogate light chain 3 has a higher expression response to IBV vaccination, which is significant in the B cell maturation and plays a crucial role in Pre-B cell receptor formation [35,36]. The high expression of these genes represented the accumulation of B cells within the spleen in response to vaccination compared with the control group.

GO and KEGG database analyses between vaccinated and control groups of TCCs (Appendix A) showed significant pathways related to inflammation and pathogen recognition in response to vaccination. The remarkable enriched pathways from the KEGG database were those involving cytokine–cytokine receptor interaction, nucleotide oligomerization domain (NOD)-like receptor signaling pathway, and TLR signaling pathway. By identifying pathogen-associated molecular pattern (PAMP), NOD-like and Toll-like receptor signaling pathways are the important connector between innate and adaptive immune responses. Additionally, from the analysis from GO database, vaccinated TCCs were highly enriched in response to chemicals, response to hormones, taxis, and cellular response to peptide. These biological processes are essential mechanisms for the host to respond to external stimuli and initiate the adaptive immune response. In addition, hormone and taxis pathways are related to the attraction and accumulation of immune cells to the spleen, allowing T and B cell response to antigens in the circulatory system [37].

### 4.2. Immune Responsiveness in White Leghorn Chickens

WLCs showed only 49 upregulated genes and 72 downregulated genes in response to vaccination (Table 2). The significantly upregulated immune-related genes were involved in the innate immune response and activation of adaptive immune responses, including IL-6 and IL-4I1. IL-6 is a crucial multifunctional cytokine related to the proinflammatory process and the activation and differentiation of T and B lymphocytes. A study found that IL-6 is crucial for mice survival after infection with influenza through the optimization of T cell regulation and the migration and phagocytic activities of macrophages [38]. However, the presence of IL-6 and TGF-β are also related to immune tolerance by promoting the expression of regulatory T cells [39]. In human studies, IL-4I1 plays an important role in the limitation of side effects from adaptive immune response by inhibiting IFN -γ production and differentiation of effector T cells and turning naive T cells into regulatory T cells [40].

According to the GO and KEGG databases, the important GO pathways in WLCs (Appendix A) were those involving the regulation of IL-6 production, IL-6 production, positive regulation of T cell proliferation, and B cell activation. The main gene involved in these enriched pathways was IL-6. This evidence stated the important role of IL-6 in immune response modulation against IBV vaccination in WLCs. For the KEGG pathway analysis (Appendix A), the significant enriched pathways were involved in the inflammatory process, lymphocyte proliferation and differentiation, including cytokine–cytokine receptor interaction, transforming growth factor-beta signaling pathway, and phagosome.

From human study, the transforming growth factor-beta (TGF- β) signaling pathway plays a critical role on the divergent function of T cell. On the one hand, TGF-β plays a role in supporting the activation of functional T cells by promoting chemotaxis of immune cells. TGF-β can induce chemotaxis of CD4+ T cells towards the CXCL12 and promote the migration of antigen-presenting cells towards lymphatic vessels and lymph nodes [41,42]. On the other hand, TGF-β also controls the immune tolerance by inhibiting T-cell receptor signaling and promoting the differentiation and function of the regulatory T cells (Tregs) [39]. The highly expressed IL-4I1 and enriched transforming growth factor-beta signaling pathway in WLCs could limit the side effects from overstimulation of the adaptive immune response. However, lower levels of effective T cells could also reduce the host’s capability to eradicate the invasive pathogens. A previous study indicated that high expression of regulatory T cells is related to susceptible traits in chicken against Marek’s infection [43].

### 4.3. Differences of Immune Responsiveness between Breeds

To investigate the effect of genetics between breeds, the control groups between TCCs and WLCs were compared. TCCs had highly expressed cytokine-related genes involved in immunological tolerance, including IL-4I1, IL-6, and IL-22. As previously described, IL-4I1 plays a significant role in the limitation of immunopathology mediators by controlling the effector Th1 and Th17 cells [44]. Additionally, IL-22 is secreted by activated dendritic cell and Th 17 cells. It functions as a proinflammatory and regenerative factor and is crucial for the protection and regeneration of barrier organs such as the lungs and gastrointestinal system [45]. In addition, immunoglobulin superfamily member 1-like 6, MHC class I polypeptide-related sequence A, and class I histocompatibility antigen, F10 alpha chain-like were enriched. An immunoglobulin superfamily member, recognized as an adhesion molecule, plays a crucial role in the mediation of cell surface interaction and pathogen perception [46]. MHC class I polypeptide-related sequence A, class I histocompatibility antigen, and F10 alpha chain-like are related to the antigen presentation process through MHC class I.

The upregulated genes in unvaccinated WLCs were those related to T cell-interacting, activating receptor on myeloid cell protein 1-like, and nuclear factor of activated T cells (Table 3). T cell-interacting, activating receptor on myeloid cells protein 1 is a triggering receptor on macrophages and neutrophils related to TNF-α and IL-8 production against pathogen infection [47]. Likewise, the nuclear factor of activated T cells is a transcriptional factor that involves many normal body processes and is well known as a crucial player in T cell activation and the determination of the fate and function of the T cell population [48]. Moreover, WLCs also highly expressed interleukin 1 receptor-like 2 and MAPK10. The main function of interleukin 1 receptor-like 2 is inhibiting IL-1 activity, which can inhibit the excess production of proinflammatory cytokines.

Comparison between vaccinated groups of the two breeds (Table 3) revealed that the genes highly expressed in TCCs were those encoding IL-8L1, C-type lectin domain family 2 member L-like, IgG Fc-binding protein-like, C-type lectin domain family 2 member L-like, class I histocompatibility antigen, F10 alpha chain-like 3, class I histocompatibility antigen, F10 alpha chain-like, MHC class I polypeptide-related sequence A, and MHC class II beta chain BLB1 response to vaccination. The IL-8L1 plays a crucial role in neutrophil activation through the promotion of cell adhesion, transendothelial migration, and killing process. Furthermore, the function of the IgG Fc-binding protein is the modulation of the adaptive immune response through fusion with specific antigenic proteins [49]. Higher expression of the above genes in the TCCs in comparison to WLCs shows the role of these genes in activation of B and T cells in subsequent IBV vaccination.

Among the vaccinated groups, WLCs showed high expression of IL-6, IL-22 receptor subunit alpha 2, T cell surface glycoprotein CD8 alpha chain-like, granzyme K, MHC class I antigen YF5, and MHC class II beta chain BLB2. The high expression of these genes in WLCs relates to T cell development and activation process responding to IBV vaccination. The T cell surface glycoprotein CD8 alpha chain-like was related to CD8 T cell accumulation and T cell activation within the spleen. In addition, granzyme K was secreted by activated macrophage, natural killer cell, and cytotoxic T lymphocyte, which functionally promotes the cytotoxicity of invader and infected cells [50]. The higher expression of these genes compared with TCCs suggests that T cell development and activation were among the major immune processes in the spleen responding to vaccination in WLCs.

From the KEGG pathway analysis (Table 5), TCCs and WLCs shared several immune-related pathways responding to vaccination, including cell adhesion molecules (CAMs), cytokine–cytokine receptor interaction, toll-like receptor (TLR) signaling pathway, salmonella infection, and influenza A. Interestingly, the gene members in each pathway were expressed differently between the two breeds of chicken. The major gene members in TCCs were IL-8L1 and BLB1, whereas the main members in WLCs were IL-6 and BLB2. BLB1 and BLB2 are class II MHC genes that are highly polymorphic and responsible for transducing signals during the B cell activation process. The results in Table 1 show that TCCs had a higher capability to produce more immunoglobulin in response to IBV vaccination compared with WLCs. Based on this experiment, we could assume that the different expression of MHC class II between TCCs and WLCs might play a significant role on varying antibody production levels against IBV vaccination. However, further investigation is needed to confirm these findings.

## 5. Conclusions

This study provides a comprehensive understanding of immune related genes and biological pathways against IBV vaccination of TCCs and commercial WLCs. On day 7 post inoculation, the vaccinated TCCs group exhibited higher levels of antibody production compared to the control group. Additionally, a set of differentially expressed genes related to innate immune responses, as well as B cell proliferation and development, were highly expressed. These genes included MAPK10, macrophage mannose receptor 1-like 2, Gallus gallus immunoglobulin-like receptor CHIR-B2-like, and V-set pre-B cell surrogate light chain 3. On the contrary, the significant enriched immune-related genes in vaccinated WLCs were IL-6, IL-4I1, and IL22RA2, which related to proinflammatory process and the activation and differentiation of T and B lymphocytes. In addition, the data analysis revealed that significant biological pathways of both two breeds were related through cell adhesion molecules (CAMs), cytokine–cytokine receptor interaction, toll-like receptor (TLR) signaling pathway and influenza A against live attenuated IBV vaccination.

## Figures and Tables

**Figure 1 animals-13-01642-f001:**
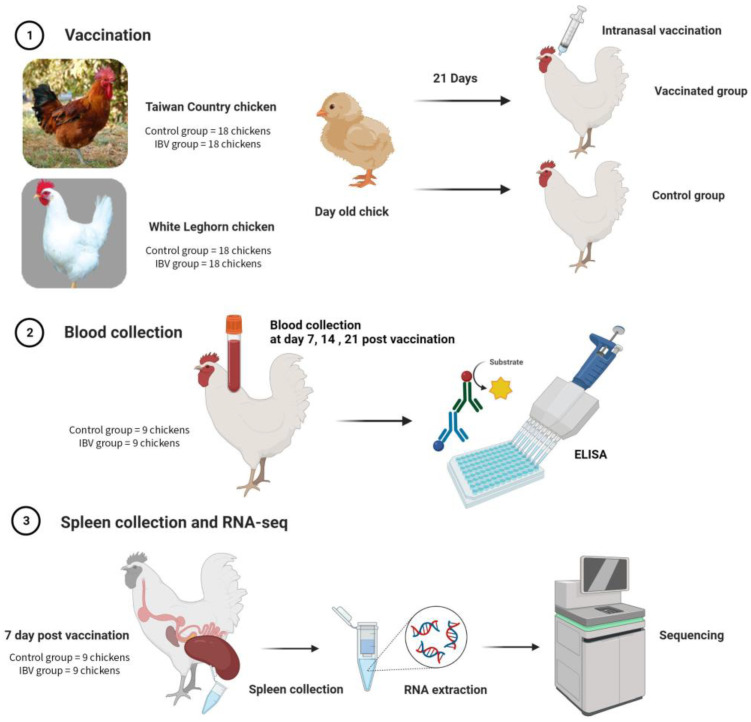
The sample collection and experimental processes of Taiwan Country chicken and White Leghorn chicken. Created with BioRender.com accessed on 9 April 2023.

**Figure 2 animals-13-01642-f002:**
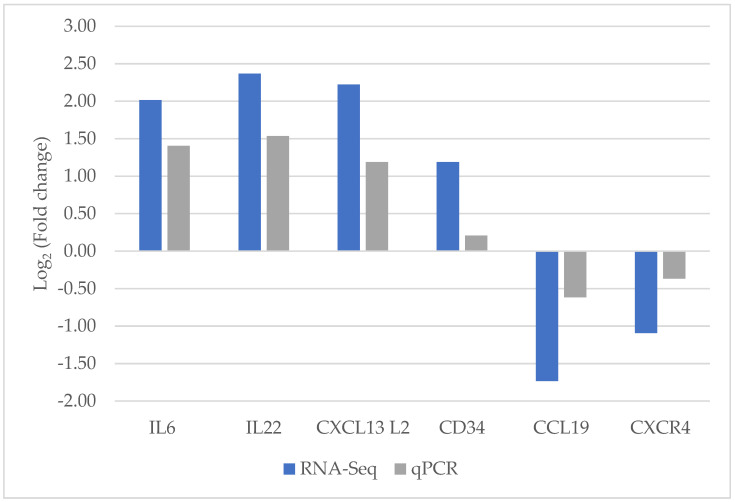
Validation of selected genes between ribonucleic acid-sequencing and quantitative real-time polymerase chain reaction by Log_2_ Fold Change.

**Table 1 animals-13-01642-t001:** Mean anti-IBV ELISA antibody titers [S (OD sample-OD negative control)/p (OD positive control-OD negative control) ratio] between the treatment and control groups of each breed of chickens.

Day	TCC		WLC	
	Treatment	Control	Treatment	Control
Day 0	1.78 ± 0.70	1.37 ± 0.36	1.72 ± 0.33	1.44 ± 0.28
Day 7	3.83 ^a^ ± 1.15	1.04 ^b^ ± 0.13	1.68 ± 0.37	1.32 ± 0.37
Day 14	31.40 ^a,x^ ± 6.55	1.04 ^b^ ± 0.37	7.90 ^a,y^ ± 1.36	0.88 ^b^ ± 0.21
Day 21	38.00 ^a,x^ ± 5.73	0.93 ^b^ ± 0.35	8.79 ^a,y^ ± 1.43	1.02 ^b^ ± 0.17

^a,b^ Mean ± SE within a breed for a given stage with different superscripts indicating different significance. ^x,y^ indicated significant differences between the breeds in a given stage (*p* < 0.05).

**Table 2 animals-13-01642-t002:** List of significant immune-related differentially expressed genes between vaccinated and unvaccinated groups within a breed at day 7 post vaccination.

Breed	Upregulated ^X^	Downregulated ^Y^	Immune Related Genes	log2FoldChange	*p*-Value
TCC	257	186	Immunoglobulin superfamily DCC subclass member 4 ^X^	1.89	3.50 × 10^−5^
			Transforming growth factor beta receptor 3 ^x^	1.43	4.82 × 10^−4^
			Macrophage mannose receptor 1-like 2 ^X^	1.40	1.60 × 10^−3^
			Mitogen-activated protein kinase 10 ^X^	1.80	5.88 × 10^−3^
			Major histocompatibility complex, class I ^X^	2.49	5.66 × 10^−3^
			V-set pre-B cell surrogate light chain 3 ^X^	2.49	7.56 × 10^−4^
			Interleukin 6 ^Y^	−2.86	4.42 × 10^−3^
			Interleukin 4 induced 1 ^Y^	−2.65	9.62 × 10^−3^
			Interleukin 22 ^Y^	−2.54	3.50 × 10^−3^
WLC	49	72	Interleukin 4 induced 1 ^X^	1.45	4.18 × 10^−5^
			Interleukin 22 receptor subunit alpha 2 ^X^	1.26	8.11 × 10^−4^
			Interleukin 6 ^X^	2.02	5.26 × 10^−3^
			MHC class I polypeptide-related sequence A ^Y^	−1.56	9.59 × 10^−6^
			Eosinophil peroxidase ^Y^	−1.36	4.65 × 10^−4^
			Class I histocompatibility antigen, F10 alpha chain-like 3 ^Y^	−1.17	3.06 × 10^−3^

^X^ Upregulated due to *p*-value ≤ 0.05; ^Y^ Downregulated due to *p*-value ≤ 0.05.

**Table 3 animals-13-01642-t003:** The list of significant immune-related differentially expressed genes between Taiwan Country chicken and White Leghorn chicken at day 7 post vaccination according to the challenge status.

Challenge Status	Breed		Immune Related Genes	log2FoldChange	*p*-Value
	TCC	WLC			
Unvaccinated	577	1118	C-X-C, motif chemokine ligand 14 ^X^	6.46	4.74 × 10^−7^
			Interleukin 22 ^X^	4.62	1.05 × 10^−4^
			Interleukin 4 induced 1 ^X^	3.49	1.25 × 10^−4^
			Interleukin 6 ^X^	3.20	1.39 × 10^−4^
			Leukocyte immunoglobulin-like receptor subfamily B member 5 ^X^	4.98	2.63 × 10^−3^
			Interleukin 1 receptor-like 2 ^Y^	−3.09	2.52 × 10^−13^
			TNF receptor superfamily member 19 ^Y^	−1.58	3.14 × 10^−8^
			T cell-interacting, activating receptor on myeloid cells protein 1-like ^Y^	−3.20	3.95 × 10^−7^
			Mitogen-activated protein kinase 10 ^Y^	−1.82	3.49 × 10^−03^
Vaccinated	470	416	Class I histocompatibility antigen, F10 alpha chain-like ^X^	6.82	4.78 × 10^−83^
			C-type lectin domain family 2 member L-like ^X^	3.78	3.35 × 10^−25^
			Major histocompatibility complex class II beta chain BLB1 ^X^	1.42	1.14 × 10^−21^
			IgG Fc-binding protein-like ^X^	1.49	4.84 × 10^−15^
			Interleukin 8-like 1 ^X^	1.54	1.03 × 10^−3^
			Major histocompatibility complex class II beta chain BLB2 ^Y^	−1.07	1.61 × 10^−13^
			T cell surface glycoprotein CD8 alpha chain-like ^Y^	−2.20	8.57 × 10^−7^
			Granzyme K ^Y^	−1.07	1.00 × 10^−7^
			MHC class I antigen YF5 ^Y^	−1.03	4.29 × 10^−6^
			Interleukin 22 receptor subunit alpha 2 ^Y^	−1.51	9.32 × 10^−5^
			Interleukin 6 ^Y^	−1.67	2.56 × 10^−2^

^X^ Upregulated due to *p*-value ≤ 0.05; ^Y^ Downregulated due to *p*-value ≤ 0.05.

**Table 4 animals-13-01642-t004:** Comparison of significant pathways from GO between the vaccinated groups of Taiwan Country chicken and White Leghorn chicken at day 7 post vaccination.

Breed	Description	*p*-Value	Count	Genes ID
TCC	Cell-cell adhesion via plasma-membrane adhesion molecules	3.60 × 10^−4^	5	CDH3/DSCAML1/FLRT3/NFASC/SDK2
	Cell differentiation	1.00 × 10^−3^	19	ACE/ZP3/OLFM1/RNASE6/ACVR2B/BRINP1/ APOA1/DSCAML1/FLRT3/TLL1/RARB/MGP/ TAGLN/SERPINB10B/CSRP2/NFASC/SDK2/MSTN
	Chemotaxis	0.164	3	FLRT3/NFASC/MSTN
	Regulation of leukocyte migration	0.214	1	MSTN
	Myeloid cell differentiation	2.29 × 10^−4^	2	ACE/SERPINB10B
WLC	Cell proliferation	7.00 × 10^−3^	7	NPPC/CDH13/ANKRD1/HPGDS/IL6/NTRK2/INHBA
	Acute inflammatory response	7.00 × 10^−3^	2	FN1/IL6
	Cell differentiation	1.10 × 10^−2^	11	NPPC/ANKRD1/VSIG1/TENM1/IL6/RPE65/THRB/NTRK2/INHBA/STMN2/BHLHE22
	Regulation of cell proliferation	1.10 × 10^−2^	6	CDH13/ANKRD1/HPGDS/IL6/NTRK2/INHBA
	Positive regulation of cell adhesion	5.20 × 10^−2^	2	CDH13/IL6
	Cell-cell adhesion via plasma-membrane adhesion molecules	5.20 × 10^−2^	2	CDH13/TENM1

**Table 5 animals-13-01642-t005:** Comparison of significant pathways from KEGG between the vaccinated groups of Taiwan Country chicken and White Leghorn chicken at day 7 post vaccination.

Breed	Description	*p*-Value	Count	Symbol
TCC	Cell adhesion molecules (CAMs)	2.40 × 10^−2^	6	BLB1/CDH3/CDH1/NRXN1/NFASC/CNTNAP2
	Cytokine-cytokine receptor interaction	3.00 × 10^−2^	8	GH/IL11RA/ACVR2B/IL8L1/BMP6/EDAR/CCR10/MSTN
	Phagosome	9.70 × 10^−2^	5	BLB1/LOC425049/MMR1L1/ATP6V0D2/TUBA1C
	NOD-like receptor signaling pathway	0.202	4	IL8SL1/CAMP/GBP/GBP4L
	Salmonella infection	0.720	1	IL8L1
	Influenza A	0.728	2	BLB1/IL8L1
	Toll-like receptor signaling pathway	0.793	1	IL8L1
	Herpes simplex virus 1 infection	0.941	1	BLB1
WLC	Cytokine-cytokine receptor interaction	6.70 × 10^−2^	5	TNFSF15/IL6/IL18RAP/IL1RL2/INHBA
	Cell adhesion molecules (CAMs)	0.148	3	NCAM2/NRXN3/BLB2
	NOD-like receptor signaling pathway	0.402	2	IL6/CASR
	Influenza A	0.448	2	IL6/BLB2
	Herpes simplex virus 1 infection	0.492	2	IL6/BLB2
	Salmonella infection	0.529	1	IL6
	Toll-like receptor signaling pathway	0.606	1	IL6

## Data Availability

Data are available for fair use to support further scientific research upon request to the corresponding author.

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
