# Peer review of "Comparison of Immune-Related Gene Expression in Two Chicken Breeds Following Infectious Bronchitis Virus Vaccination"

_animals, 2023, doi:10.3390/ani13101642_

Round 1

Reviewer 1 Report

Over all I thought the manuscript was well written. 

Conclusions

Based on my interpretation of the conclusions the strains responsed differently in their respective immune response.  Could it be said that even though the IBV response patways were different their immune response to IBV could be considered equivalent or would one be considered better than the other?

Author Response

Dear Reviewer 1,

I hope this letter finds you well. I am writing to express my sincere gratitude for taking the time to review my manuscript titled “Comparison of Immune-Related Gene Expression in Two Chicken Breeds following Infectious Bronchitis Virus Vaccination”. I appreciate your insightful comments and constructive criticism, which have helped me improve the quality of my research article significantly.

Please find attached a point-by-point response to your comments. I hope that you find the changes made to the manuscript satisfactory and that the revised version meets your expectations.

Once again, I would like to express my sincere gratitude for your invaluable feedback and constructive criticism. Your comments have significantly contributed to the improvement of the quality of my research article. I look forward to hearing from you soon regarding the revised manuscript.

Thank you for your time and consideration.

Yours sincerely,

  Schwann

Reviewer 2 Report

Animals Editorial Office (Mr. Alex Tao):

I have reviewed this paper (Manuscript ID: animals-2287908). The experimental design of this paper was reasonable and the idea was simple and clear. After immunizing two different breeds of chickens, the dynamic changes of immune-related genes and related signaling pathways were compared. The use and comparative analysis of these two genetically distinct breeds of chickens have potential to elucidate the host genetic components that are associated with differential susceptibility to infections. Moreover, findings of this study would increase the availability of effective disease control strategies. Although the experiment and analysis of the paper are good, there are still some small details to be corrected.

1. In lines 173-175, the meaning of the text is unclear.

2. In table S5-S6, the expression of P values needs to be simplified.

3. In line 276, format needs to be modified.

4. In lines 306-307, this content can be put in 4.3.

5. In lines 443-444, the meaning of this sentence is unclear and needs to be revised.

6. In lines 488-490, the format of reference 10 needs to be reviewed.

Therefore, I suggest that the author review the sentences and grammar of the whole article to make sure that the meaning of the article is clear.

Author Response

Dear Reviewer 2,

I hope this letter finds you well. I am writing to express my sincere gratitude for taking the time to review my manuscript titled “Comparison of Immune-Related Gene Expression in Two Chicken Breeds following Infectious Bronchitis Virus Vaccination”. I appreciate your insightful comments and constructive criticism, which have helped me improve the quality of my research article significantly.

Please find attached a point-by-point response to your comments. I hope that you find the changes made to the manuscript satisfactory and that the revised version meets your expectations.

Once again, I would like to express my sincere gratitude for your invaluable feedback and constructive criticism. Your comments have significantly contributed to the improvement of the quality of my research article. I look forward to hearing from you soon regarding the revised manuscript.

Thank you for your time and consideration.

Yours sincerely,

  Schwann

Reviewer 3 Report

Animals- 2287908

 Comparison of Immune-Related Gene Expression in Two 2 Chicken Breeds following Infectious Bronchitis Virus Vaccination

The work aims to identify differences in the expression of immune response-associated genes between two strains of chickens after immunization with an attenuated vaccine. By applying a transcriptomic analysis, the authors identified several differentially expressed genes related to different immune response pathways. The research provides relevant data on the influence of strain on the response to vaccination.

Comments

-          The manuscript presents some stylistic problems that make its reading and interpretation confusing.

Materials and Methods:

-          The distribution of animals among the different experiments is not clear. Each group has 27 chickens, nine were used to study the humoral immune response up to 21 days post-vaccination. Another nine were sacrificed on day 7 post-vaccination for RNA extraction from the spleen. What were the other nine used for?

-          Which is the adjuvant of the vaccine? This information should be added and considered in the discussion about the expression of innate response-related genes.

Results:

-          How are the antibody titers expressed? Please define the S/P ratio

-          Tables S4, 3: Change uninfected by unvaccinated, infected by vaccinated

-          Tables 4, 5: Change column titles p-value and count.

-          How was the gene expression compared in unvaccinated animals? TWC/WLH or WLH/TWC? (Tables 3, S4) The presentation of the results are not clear.

Discussion:

-          The discussion should be rewritten to integrate the results and not describe only gene functions. The relationship of the findings to the pathology and to the protective ability of the vaccine is missing.

-          Some concepts should be revised, eg.

Lines 365-367: TGF-β critically supports functional T cell production by regulating chemotaxis and activating the cellular responses through immune cells, such as lymphocyte, natural killer cells, dendritic cell and macrophage

The sentence does not focus on TGF-b functions or target cells

Lines 368-369: TGF-β also controls the immune tolerance by inhibiting T-cell receptor signaling and inducting the differentiation of Th 17 cells

TGF-b-induced tolerance is not mediated by Th17 cells.

In the paragraph starting on line 376: 4.3. Differences of immune responsiveness between breeds, the comparison does not take into account that unvaccinated animals received adjuvants. Therefore, the pattern of gene expression may be influenced by this stimulus.

Line 411: Therefore, the enormously expressed IgG Fc-binding protein-like indicates the preferential expression of humoral immunity of TWC chickens.

It is difficult to see the “enormously” increase in the expression of this protein. It is also risky to attribute the higher antibody levels observed in TWC chickens to the expression of this protein.

Line 432: Previous studies have revealed the effect of MHC polymorphism on variations of antibody production level and susceptibility traits against viral infection

Which is the relationship of the polymorphism level with the results obtained in this work?

Line 440: This study provided a comprehensive understanding of immune related genes and biological pathways against IBV infection…

The study was done on vaccinated animals, then conclusions on the response to infection are not adequate.

Author Response

Dear Reviewer 3,

I hope this letter finds you well. I am writing to express my sincere gratitude for taking the time to review my manuscript titled “Comparison of Immune-Related Gene Expression in Two Chicken Breeds following Infectious Bronchitis Virus Vaccination”. I appreciate your insightful comments and constructive criticism, which have helped me improve the quality of my research article significantly.

Please find attached a point-by-point response to your comments. I hope that you find the changes made to the manuscript satisfactory and that the revised version meets your expectations.

Once again, I would like to express my sincere gratitude for your invaluable feedback and constructive criticism. Your comments have significantly contributed to the improvement of the quality of my research article. I look forward to hearing from you soon regarding the revised manuscript.

Thank you for your time and consideration.

Yours sincerely,

  Schwann

Round 2

Reviewer 2 Report

NO

Author Response

Dear Reviewer 2,

Thank you so much for reviewing our manuscript. We greatly appreciate the time and effort you took to provide us with insightful feedback and valuable suggestions, which helped us to enhance the quality of our work. Your expertise in the field was truly invaluable, and we are grateful for the opportunity to benefit from it.

Thank you once again for your assistance.

Yours sincerely,

   Schwann

Reviewer 3 Report

The questions and suggestions were adequately answered by the authors.

Author Response

Dear Reviewer 3,

Thank you so much for reviewing our manuscript. We greatly appreciate the time and effort you took to provide us with insightful feedback and valuable suggestions, which helped us to enhance the quality of our work. Your expertise in the field was truly invaluable, and we are grateful for the opportunity to benefit from it.

Thank you once again for your assistance.

Yours sincerely,

   Schwann